# A synthetic enzyme built from DNA flips $10^7$ lipids per second in biological membranes

Alexander Ohmann [1], Chen-Yu Li [2], Christopher Maffeo[3], Kareem Al Nahas [1], Kevin N. Baumann [1], Kerstin Göpfrich [1], Jejoong Yoo [3], Ulrich F. Keyser [1] & Aleksei Aksimentiev [2,3,4]

Mimicking enzyme function and increasing performance of naturally evolved proteins is one of the most challenging and intriguing aims of nanoscience. Here, we employ DNA nanotechnology to design a synthetic enzyme that substantially outperforms its biological archetypes. Consisting of only eight strands, our DNA nanostructure spontaneously inserts into biological membranes by forming a toroidal pore that connects the membrane's inner and outer leaflets. The membrane insertion catalyzes spontaneous transport of lipid molecules between the bilayer leaflets, rapidly equilibrating the lipid composition. Through a combination of microscopic simulations and fluorescence microscopy we find the lipid transport rate catalyzed by the DNA nanostructure exceeds $10^7$ molecules per second, which is three orders of magnitude higher than the rate of lipid transport catalyzed by biological enzymes. Furthermore, we show that our DNA-based enzyme can control the composition of human cell membranes, which opens new avenues for applications of membrane-interacting DNA systems in medicine.

[1] Cavendish Laboratory, University of Cambridge, JJ Thomson Avenue, Cambridge CB3 0HE, UK. [2] Center for Biophysics and Quantitative Biology, University of Illinois at Urbana-Champaign, 1110 West Green Street, Urbana, IL 61801, USA. [3] Department of Physics, Center for the Physics of Living Cells, University of Illinois at Urbana-Champaign, 1110 West Green Street, Urbana, IL 61801, USA. [4] Department of Physics and Beckman Institute for Advanced Science and Technology, University of Illinois at Urbana-Champaign, 1110 West Green Street, Urbana, IL 61801, USA. These authors contributed equally: Alexander Ohmann, Chen-Yu Li. Correspondence and requests for materials should be addressed to U.F.K. (email: ufk20@cam.ac.uk) or to A.A. (email: aksiment@illinois.edu)

The two leaflets of a mammalian cell's plasma membrane are made up from chemically distinct mixtures of phospholipids[1]. Control over the asymmetric partitioning of phospholipids is critically important to the cell's health and functions. Thus, a loss of the asymmetry can trigger unregulated apoptosis[2] and could lead to development of Alzheimer's disease[3]. Three groups of enzymes—flippases, floppases, and scramblases[4,5]—can move lipids from one leaflet to the other. In contrast to flippases and floppases that require energy input for maintaining the asymmetric lipid composition, scramblases are activated to rapidly and passively dismantle the asymmetric partitioning of the lipids, which typically occurs during critical events such as cell activation, blood coagulation, and apoptosis[6–9]. Defects in the enzyme-catalyzed scrambling of membrane phospholipids in blood cells could hinder thrombin formation and lead to Scott syndrome[10]. Impaired lipid scrambling has been shown to weaken the immune system and evoke the autoimmune response by exposing self-antigens[11]. Thus, development of biocompatible and easy to adapt synthetic analogues to repair and/or control lipid scrambling activity in cell membranes is of considerable medical interest.

So far, membrane-spanning DNA nanostructures have emerged primarily as synthetic mimics of biological membrane channels[12–20]. Critical for lipid membrane insertion of DNA nanostructures was their decoration with hydrophobic anchors[12,14–20] as the bilayer's hydrophobic core presents a high energetic barrier for DNA[21]. Recently, we have shown that a porphyrin-modified and membrane-inserted single DNA duplex promotes formation of a toroidal water-filled pore surrounding the duplex[17]. Here, we utilize the DNA-induced toroidal pore to design a fully functional synthetic scramblase that facilitates rapid mixing of lipids between membrane leaflets. Strikingly, the scrambling activity of our de novo designed DNA-made enzyme outperforms any known biological scramblase by several orders of magnitude. This is remarkable given that the catalytic rates of previous enzymatically active DNA nanostructures fall orders of magnitude behind natural benchmarks[22,23].

## Results

**Design and assembly of DNA nanostructure**. We have designed a DNA nanostructure consisting of eight chemically synthesized DNA strands, two of which are modified with a covalently linked cholesterol group on their 3′ ends (Fig. 1; for design details and sequences see Supplementary Fig. 1 and Supplementary Table 1). Diluted in a salt buffer containing Mg²⁺-ions and following a previously described temperature annealing protocol[16], the strands self-assemble into four interconnected DNA duplexes with a designed length of ~13 nm (Fig. 1a). The hydrophobic cholesterol modifications, which are necessary to anchor the charged nanostructures into a lipid bilayer, are strategically positioned in the center of the construct pointing diagonally away from the central pore (Fig. 1b). To verify the folding and incorporation of these hydrophobic tags into the nanostructure, non-denaturing polyacrylamide gel electrophoresis (PAGE) was performed on constructs folded from either eight unmodified DNA strands, or with one or two cholesterol-modified oligonucleotides (Fig. 1d). The gel shows intensity bands that, under the same experimental conditions, shift toward shorter run lengths with every additional cholesterol moiety. The observed shifts are consistent with the increased molecular weight and cross-sections of the modified DNA nanostructures, which confirms their successful assembly and incorporation of the cholesterol tags. PAGE was performed without surfactants, demonstrating that no detergents are necessary to form our monomeric, functional constructs.

**All-atom MD simulations of DNA-induced lipid scrambling**. Having experimentally validated the feasibility of folding our cholesterol-modified DNA nanostructures, we used the all-atom molecular dynamics (MD) method to determine if the structures could induce lipid scrambling when inserted into a lipid bilayer. Following a previously described protocol[24], we built an all-atom model of the DNA nanostructure embedded in a diphytanoyl phosphatidylethanolamine (DPhPE) lipid bilayer membrane and solvated in 1 M KCl. The entire system was first equilibrated for ~230 ns with the DNA nanostructure constrained to its initial idealized conformation, allowing for lipids and water to adopt an equilibrium configuration where the lipid head groups form a toroidal pore around the nanostructure (Fig. 2a; see Supplementary Note 1 for detailed description of the simulation protocols). The system was then simulated without any constraints for ~2 μs (Fig. 2b). Comparison of the initial and the final configurations indicates that several lipids have completely transferred from one leaflet to the other (Fig. 2a, b; Supplementary Fig. 2). Visual inspection of the MD trajectory revealed diffusion of lipids along the walls of the toroidal pore. The lipids

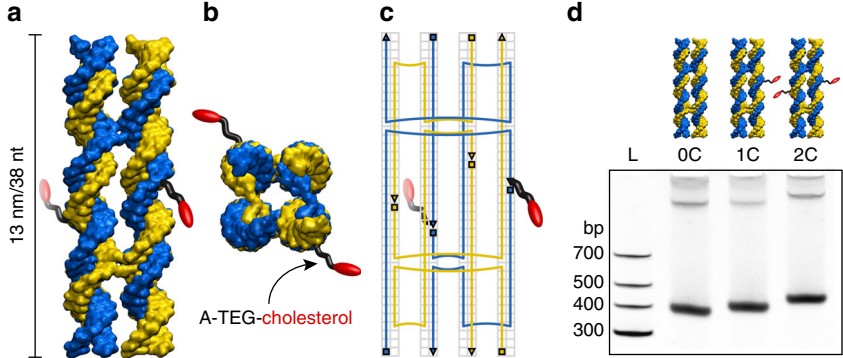

**Fig. 1** Design of the lipid-scrambling DNA nanostructure. **a** Side and **b** top view of a 3D representation of the assembled DNA nanostructure. Complementary DNA strands are displayed in blue and yellow. Cholesterol modifications (red) at two specific locations are covalently bound to the DNA via an adenine-triethyleneglycol linker (A-TEG, black). **c** 2D schematic illustrating the pathway of the DNA single strands as well as crossover and modification positions. Triangles and squares denote the 3′ and 5′ ends of the strands, and the background grid specifies locations of individual nucleotides. **d** Non-denaturing 10% PAGE of DNA nanostructures annealed without (0C), with one (1C) or with two (2C) cholesterol modifications next to a DNA molecular weight ladder (L). The highest intensity band corresponds to a major population of monomeric structures. The low intensity bands suggest the presence of small amounts of dimers and multimers

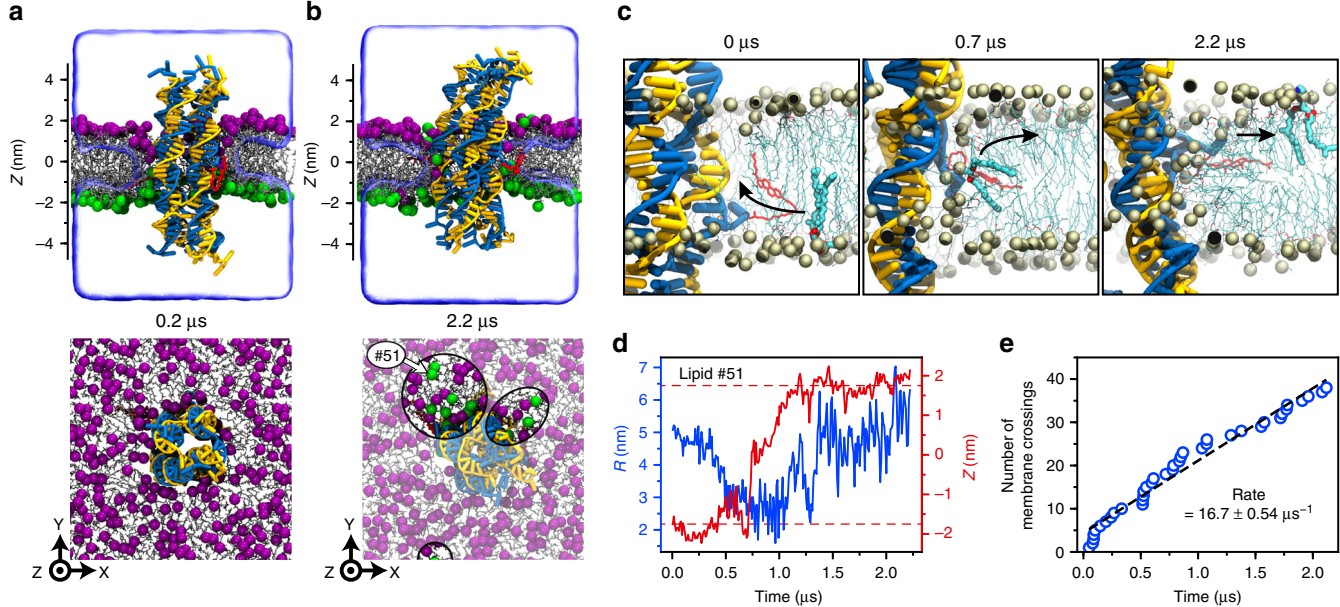

**Fig. 2** All-atom MD simulation of lipid scrambling induced by a DNA nanostructure. **a**, **b** Microscopic configuration of the simulated system at the beginning (**a**) and the end (**b**) of the free equilibration simulation. Top: cut-away view showing the DNA nanostructure (blue and yellow) embedded in a DPhPE lipid membrane (gray) via cholesterol tags (red). Lipid head groups located in the upper and lower leaflet of the bilayer at 0.2 μs are highlighted using purple and green spheres, respectively. Bottom: Top view of the simulated system; the electrolyte solution is not shown for clarity. Black ellipses highlight DPhPE head groups that resided at 0.2 μs in the lower leaflet of the bilayer. For lower leaflet lipid arrangements see Supplementary Fig. 2. **c** Sequence of microscopic conformations illustrating spontaneous inter-leaflet transfer of one lipid molecule (#51, highlighted by the bold depiction) during the 2.2 μs MD simulation. For clarity, all other lipid molecules are shown using a different representation; the electrolyte solution is not shown. Arrows illustrate the trajectory of the lipid during the simulation. **d** Radial distance relative to the center of mass of the nanostructure $R$ (left axis) and the $Z$ coordinate of the phosphorus atoms (right axis) of a lipid molecule (#51, see **b** and **c**) undergoing a complete transfer from the lower to the upper leaflet of the bilayer. Dashed red lines indicate $Z$-coordinate thresholds for the upper and lower bilayer leaflet. Data shown were sampled at 0.24 ns and averaged in 9.6 ns blocks. Supplementary Figure 3 and Supplementary Movies 2-4 show additional examples of inter-leaflet lipid transfer. **e** Cumulative number of inter-leaflet transfer events vs. simulation time. A linear fit (black dashed line) yields the average transfer rate; the error indicates the estimated standard deviation of the nonlinear least squares fit

forming the inner surface provide a continuous passage from one leaflet of the membrane to the other. The diffusive motion of individual lipid molecules was not correlated, occurred in both transport directions, and produced zero net transport of lipids from one leaflet to the other, as expected. Figure 2c and Supplementary Movies 1-4 provide examples of such inter-leaflet transfer events.

To quantitatively characterize the inter-leaflet transport of lipids, we computed the $Z$ coordinate of each lipid's phosphorus atom and its radial distance from the center of the DNA nanostructure, $R$, as a function of the simulation time (Supplementary Fig. 3). Figure 2d shows an example of a typical translocation, where a lipid molecule is seen to move from the bottom leaflet ($Z < -1.75$ nm) to the top one ($Z > 1.75$ nm), approaching the DNA nanostructure ($R < 3$ nm) during the transfer process. As the lipid molecule approaches the DNA nanostructure, it reorients itself to expose its polar head group to the DNA nanostructure. Supplementary Movie 1 illustrates this permeation trajectory. Counting an inter-leaflet transfer event each time the $Z$ coordinate of a lipid phosphorus atom changes from being less than $-1.75$ nm to being more than 1.75 nm or vice versa, we plot in Fig. 2e the cumulative number of transfer events vs. the simulation time. A linear fit yields the average transfer rate of $16.7 \pm 0.54$ lipids per μs.

A slower yet significant spontaneous transfer of lipids was observed in an additional 2 μs simulation of the same DNA nanostructure embedded in a diphytanoyl phosphatidylcholine (DPhPC) lipid bilayer membrane (Supplementary Fig. 4), suggesting that the rate of lipid transport facilitated by the same

nanostructure can be lipid-type dependent. Quantitative analysis of lipid density within the toroidal pore reveals a rather modest, of the order of 1 $k_B T$, free energy barrier to inter-leaflet transport (Supplementary Fig. 5), which is in accord with our earlier conclusion that transport of lipids occurred via diffusive motion. Given the general character of such transport, we can expect any lipid-spanning DNA nanostructures to exhibit lipid scrambling activity as long as they promote formation of a toroidal pore. Indeed, analysis of our previous MD simulations of single duplex[17] and funnel-like[18] DNA pores revealed an average inter-leaflet transport of 4 and 200 lipids per microsecond, respectively. In contrast, no lipid scrambling was observed for a DNA nanostructure that had a modified DNA backbone and did not allow the toroidal pore to form[13,24].

**Scrambling rate analysis via Brownian dynamics (BD) simulations**. To accurately determine the rate of lipid scrambling and its dependence on the pore-to-lipid ratio, we built a coarse-grained BD representation of the toroidal pore surrounding a DNA nanostructure. In our BD model, the head groups of the lipids are represented by point particles (beads) whereas the presence of all other components of the system, including the DNA nanostructure, the lipid tail, and the electrolyte solution, are modeled implicitly. The bead–bead interaction is described by a short-range repulsive potential (Supplementary Fig. 6); the diffusivity of each bead depends on its radial distance from the center of the nanostructure (Supplementary Fig. 7); a 3D potential confines the motion of the beads to the volume

accessible to the lipid head groups in all-atom simulations (Supplementary Fig. 8). Supplementary Note 2 and Supplementary Figs. 9 and 10 provide detailed descriptions of the simulation methods and their validation against the results of all-atom simulations.

Figure 3a illustrates the distribution of the lipid head groups at the beginning and after 48 μs of a BD simulation. A significant proportion of the beads migrated from one leaflet to the other, through the toroidal pore (Supplementary Movie 5 illustrates a typical BD trajectory). Figure 3b shows the $Z$ coordinate of two representative beads in BD simulations of the system containing a lipid patch of $L = 24$ nm on each side. Using the same definition of an inter-leaflet transfer event as in the analysis of the all-atom MD simulations, one can identify five transfer events in Fig. 3b. Defining the time interval between two consecutive transfer events as $\tau$ and taking all lipid head groups into account, we obtain a normalized probability of observing a transfer event, Fig. 3c. The result can be fitted by an exponential distribution $e^{(-t/\tau_0)}/\tau_0$, yielding the average transfer rate, $1/\tau_0$, of $23 \pm 1$ μs$^{-1}$.

To determine the rate of lipid scrambling $k$ from BD simulations, we count the number of lipids that have never ventured to the other leaflet as a function of simulation time and fit the resulting dependent by a single exponential function $e^{-kt}$ (Fig. 3d). As expected, the scrambling rate $k$ depends on the system size: faster scrambling is observed for smaller lipid-patch systems in the presence of the same DNA nanostructure.

In experiment, a low fraction of fluorescently labeled lipids is used as tracers to assess lipid scrambling as described below. In our simulations we mimic such selective labeling by randomly choosing 1% of all lipid heads (1 and 6 beads for $L = 12$ and

24 nm system, respectively) to represent the modified lipids. The number of labeled lipids remaining in their original membrane leaflet decreased in discrete steps (dashed lines in Fig. 3d), however, when averaged over all possible realizations, the decay curve yielded the same average scrambling rate as when all lipid trajectories were used for the analysis.

To elucidate the dependence of the lipid transfer and scrambling rates on the pore density, we repeated our BD simulation for lipid patches of various dimensions ($L = 12, 16, 20, 24,$ and 36 nm, Supplementary Table 2) containing the same toroidal pore. The lipid transfer rate, Fig. 3e, does not exhibit a strong dependence on the lipid patch size, which we characterize using the pore-to-lipid ratio $r$ computed using the average number of lipids in one of the leaflets of the membrane. The scrambling rate increases with $r$ (Fig. 3e). The following simple mathematical expression $k = r/\langle \tau_0 \rangle$, where $1/\langle \tau_0 \rangle$ is the system-size-averaged transfer rate, reproduces the simulated scrambling rate. Thus, for the range of systems studied using the BD approach, lipid diffusion toward the toroidal pore does not limit the rate of lipid scrambling.

**Lipid scrambling experiments on giant artificial vesicles.** Following the computational characterization, we experimentally measured scrambling activity using a dithionite reduction assay[25–27] adapted to giant unilamellar vesicles (GUVs) that we made via electroformation from 2-Oleoyl-1-palmitoyl-sn-glycero-3-phosphocholine (POPC) and trace amounts of phosphatidylcholine (PC) labeled with a nitrobenzoxadiazole (NBD) fluorophore (Fig. 4a). Upon addition of sodium dithionite (Na$_2$S$_2$O$_4$),

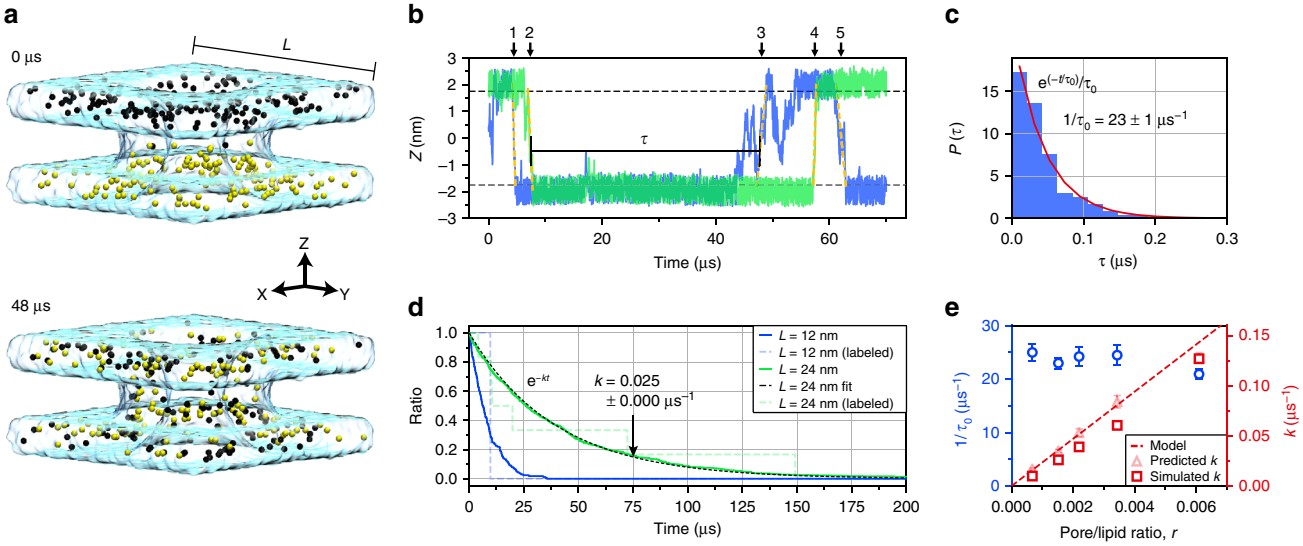

**Fig. 3** BD simulation of lipid scrambling. **a** Distribution of lipid head groups at the beginning (0 μs) and after 48 μs of BD simulation. Black and yellow spheres represent lipid head groups initially located in the upper and lower bilayer leaflets, respectively. The cyan semi-transparent surface schematically illustrates the volume accessible to lipid head groups during the simulation; $L$ denotes the size of the lipid patch (here $L = 12$ nm). **b** $Z$ coordinates of two representative lipid head groups. Horizontal dashed lines (at $Z = \pm1.75$ nm) indicate approximate boundaries of upper and lower leaflets. The traces feature five inter-leaflet transfer events; $\tau$ defines the interval between two consecutive events. Head group trajectories were sampled every 2.4 ns. **c** Normalized probability of observing an inter-leaflet transfer event within time interval $\tau$. An exponential fit (red line) yields the average transfer rate $1/\tau_0 = 23.0 \pm 1.03$ μs$^{-1}$. Data were obtained from a 500 μs trajectory of the $L = 24$ nm system sampled every 2.4 ns. **d** Fraction of lipid head groups remaining in the upper bilayer leaflet vs. time elapsed from the beginning of the simulation. Data are shown for two systems differing by the size of the lipid patch. Lipids reentering the leaflet were not included in the fraction calculation. The black dashed line shows an exponential fit to the curves; the fitting parameter $k$ is the scrambling rate. Dashed lines plot the same quantity for randomly chosen lipids mimicking experimental conditions where only a small amount of fluorescently labeled lipids is used to assess lipid scrambling. **e** Simulated transfer rate (left axis) and scrambling rate (right axis) vs. pore-to-lipid ratio. Data were derived from BD simulations of systems of different lipid patch size ($L = 12, 16, 20, 24,$ and 36 nm). Scrambling rate extracted directly from the simulation is plotted using squares. Dashed line plots $k = r/\langle \tau_0 \rangle$ curve; transparent triangles indicate $k = r/\tau_0$, where $1/\langle \tau_0 \rangle$ and $1/\tau_0$ are system-size-averaged and system-size-specific lipid transfer rates, respectively. Error bars indicate standard deviation

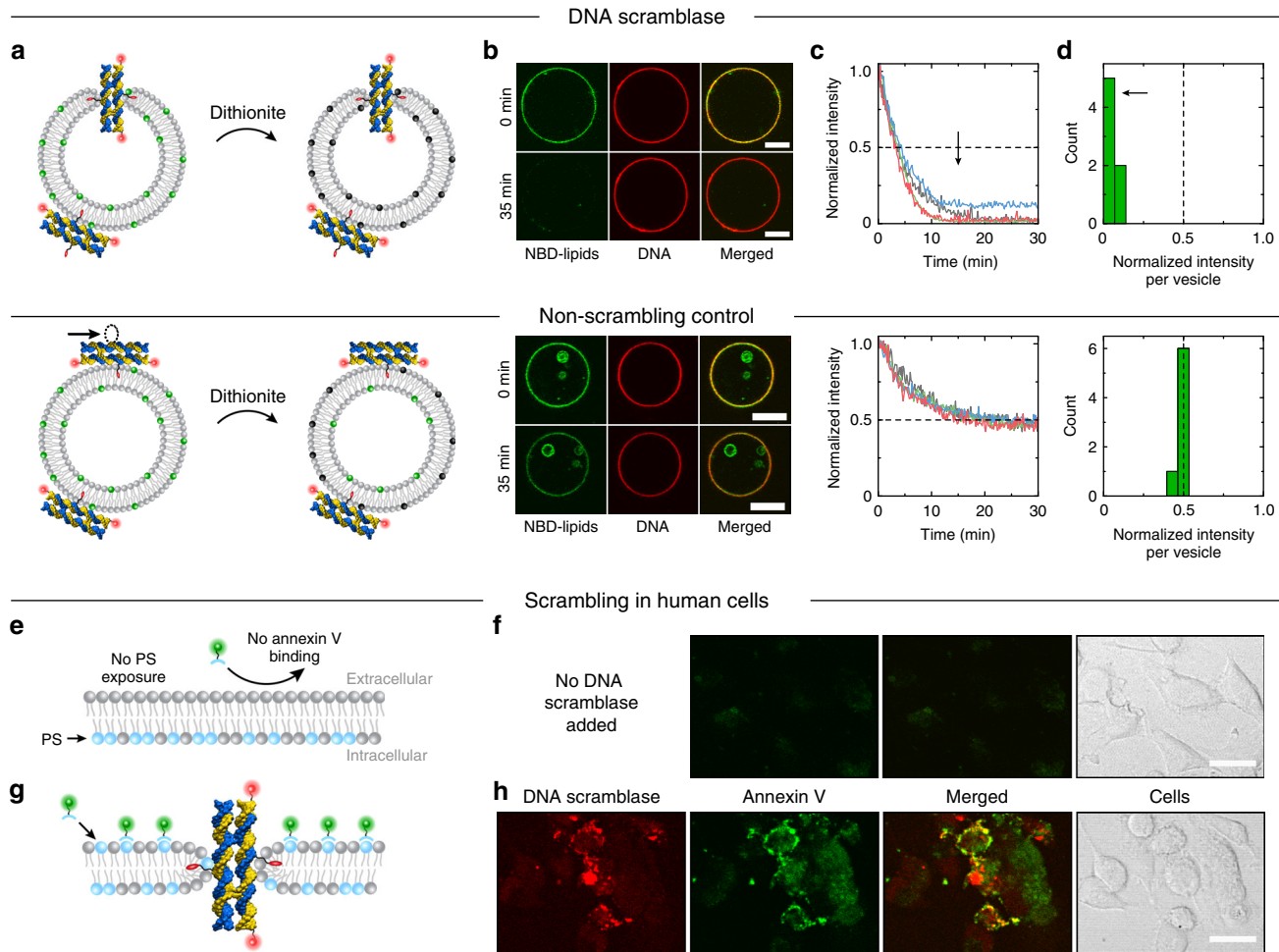

**Fig. 4** Experimental demonstration of lipid scrambling by DNA nanostructures in lipid vesicles and human cells. **a** Schematic illustration of dithionite reduction assay. POPC (gray) vesicles containing a fraction of NBD-labeled lipids (green) are incubated with DNA constructs tagged with Cy3 dye (red spheres). Upon dithionite addition, NBD fluorophores are irreversibly reduced (black). The nanostructure design containing two cholesterol modifications (2C, top) allows for membrane insertion to induce lipid scrambling whereas the constructs with one cholesterol (1C, bottom) do not. **b** Confocal fluorescence microscopy images of GUVs containing NBD fluorophores (green) and incubated with Cy3-labeled DNA nanostructures (red) showing the same vesicle before and 35 min after dithionite addition roughly at the equatorial plane. The third column displays a merged image of the red and green channels. Scale bars are 10 μm. **c** Graphs showing four representative fluorescence intensity traces of NBD fluorescence reduction over time for both 1C and 2C designs. Values have been normalized to the initial intensity per vesicle and aligned to the onset of dithionite reduction. **d** Histograms of residual NBD fluorescence intensity at 35 min after dithionite addition normalized to the initial intensity for each vesicle. Data were obtained from four (2C) and one (1C) experiments using the same GUV stock solution. Black arrows and dashed lines in **c** and **d** highlight the shifted fluorescence intensity values for 2C nanostructures in contrast to the 1C control. Data shown were collected from vesicles with a diameter above 6 μm (for smaller vesicles see Supplementary Fig. 12). **e**, **g** Schematic illustration of FITC-labeled annexin V binding assay without (**e**) and with membrane-inserted DNA nanostructures (**g**). Phosphatidylserine (PS) lipids are highlighted in light blue. **f**, **h** Confocal microscopy images of fixed cells after incubation with DNA folding buffer only (**f**) or with Cy3-labeled 2C DNA scramblases (**h**). While no annexin V binding occurs in the negative control, incubation with DNA scramblases showed cell attachment of the DNA nanostructures (red) associated with increased annexin V (green) binding (see merged signal of both channels) due to PS exposure. Corresponding bright field images illustrate cell locations (scale bar is 20 μm)

the membrane-impermeable anion dithionite ($[S_2O_4]^{2-}$) reduces the NBD fluorophores irreversibly. If no lipid scrambling occurs, only NBD in the outer leaflet of the vesicle membrane is bleached, effectively reducing the fluorescence intensity to 0.5 of its initial value. If our DNA nanostructure is present and inserted, lipids can migrate from the inner leaflet to the outer where the NBD dyes would be reduced by dithionite resulting in the decrease of fluorescence below 0.5 over time (Fig. 4a top).

POPC lipids are ideally suited for dithionite reduction assays as they show negligible rates of spontaneous flip–flop in reconstituted vesicles (half times > 1000 h)[28] and their fatty acid tails are the most abundant in naturally occurring lipid mixtures[28], classifying them particularly representative for

biological membranes. Furthermore, POPC lipids minimize the differences in lipid tail chemistry between the vesicle-forming lipids and the NBD-labeled PC tracer lipids making the tracer lipids a more accurate representation of the bulk mixture. Ultimately, the advantage of using GUVs is that they can be observed via fluorescence microscopy, which allowed us to directly verify lipid scrambling at the single vesicle level. At the same time, we were able to confirm the correlation of scrambling with the design and attachment of our DNA nanostructures.

For the experiment, the vesicles were incubated in a microscope chamber in the presence of 100 nM of folded 2C DNA nanostructures at a physiological pH of 7.4 and left to settle down due to a density gradient between intravesicular sucrose

and extravesicular glucose. The design of the DNA nanostructures was identical to the simulated model apart from added Cy3-labels that enabled fluorescence visualization (Fig. 4a top and Supplementary Fig. 1a). After incubation, vesicles were imaged using confocal fluorescence microscopy. A fluorescent ring was observed in both the NBD and the DNA channel (Fig. 4b top). Their co-localization (see merged image in Fig. 4b) demonstrates successful attachment of DNA nanostructures to the lipid vesicles[16–18]. After focusing on one field of view and establishing the initial intensity of NBD and DNA signals separately, dithionite solution was added (4.5 mM final concentration) while recording both channels over time. Care was taken not to move vesicles during the dithionite addition, and the buffer conditions were optimized to avoid significant osmotic pressure (Supplementary Fig. 11 and Supplementary Table 3). Figure 4b shows fluorescence images of one vesicle incubated with 2C nanostructures taken before and approximately 35 min after dithionite addition. The NBD signal has completely vanished suggesting that the DNA nanostructures successfully induced lipid scrambling so that inner-leaflet NBD-lipids could migrate to the outer layer where they were reduced by dithionite extinguishing any NBD fluorescence. As the DNA signal was unaffected by dithionite and the nanostructures still remained attached, the fluorescence intensity in the DNA channel could be used to localize the vesicle membrane despite the complete loss in fluorescence in the NBD channel. This enabled the acquisition of intensity traces of different vesicles over time, all showing an exponential decrease in fluorescence to almost zero (Fig. 4c top). About half an hour after dithionite addition all larger vesicles ($d > 6\,\mu m$) displayed a reduction in fluorescence of over 87% with an average residual fluorescence of only $\sim 5 \pm 4\%$ ($n = 7$, Fig. 4d top). Thus, our designed DNA nanostructures can induce lipid scrambling in biological membranes. Smaller vesicles ($d < 6\,\mu m$) also showed a significant intensity reduction (Supplementary Fig. 12a). Over all experiments, more than 85% of vesicles incubated with the 2C nanostructures showed an intensity reduction indicative of lipid scrambling.

As a straightforward control experiment, we employed the same DNA nanostructure containing only a single cholesterol tag (1C, Fig. 4a bottom). Previously, it has been shown experimentally[16,29] that one cholesterol tag is not sufficient to produce membrane insertion of similar-sized DNA nanostructures. Therefore, while this control construct would still attach to the lipid vesicles, membrane insertion and lipid scrambling are excluded with no other difference in experimental conditions. Similar to the 2C design, incubation of vesicles in the presence of 1C nanostructures resulted in a fluorescent ring in the DNA channel before and after dithionite addition (Fig. 4b bottom) indicating membrane attachment. However, the NBD fluorescence intensity after 35 min remained at about 50% indicating the absence of scrambling activity. Example fluorescence intensity traces (Fig. 4c bottom) show the expected exponential decrease plateauing at $48 \pm 1\%$ ($n = 7$, Fig. 4d bottom and Supplementary Fig. 12b). Thus, only the outer-leaflet NBD-labeled lipids have been reduced by dithionite as the 1C DNA nanostructures could not insert into the membrane and induce lipid scrambling. Another independent set of experiments for the dithionite reduction assay confirmed the difference in the scrambling activity of the 1C and 2C designs (see Supplementary Note 4, Supplementary Table 4, and Supplementary Fig. 13).

As DNA is negatively charged, permeation of anions through DNA-induced lipid pores is expected to be much slower than cation permeation due to electrostatic repulsion. We previously reported simulations of a larger, membrane-inserted DNA nanostructure that showed significantly decreased $Cl^-$ ion over $K^+$ ion permeation[24]. In accordance with these results, the performed all-atom simulations on our DNA scramblase design similarly reveal a 93% reduction of $Cl^-$ ion permeation compared to that of $K^+$ ions. In the dithionite reduction assay, the NBD-reducing dithionite anion $[S_2O_4]^{2-}$ is larger than $Cl^-$ ions and, most importantly, it is twice negatively charged. Therefore, the negative charge of the DNA nanostructure, in combination with an overall low ionic strength of the buffer solution used in the experiments, is expected to present a barrier to dithionite permeation through the toroidal pore.

To demonstrate the low permeation rate of dithionite through the DNA-induced toroidal pore, we performed a series of control experiments. According to a previously described protocol[30], we synthesized a fluorescent probe from NBD and a 24-unit polyethylene glycol (NBD-PEG; Supplementary Fig. 14a), and encapsulated it in POPC vesicles. The covalent attachment of the PEG chain was expected to increase the hydrodynamic radius of the NBD dye, preventing its direct permeation through the membrane and through DNA-induced toroidal pores. After incubation of these vesicles with our 2C DNA scramblases, NBD dyes inside the vesicles were photobleached. Measurements of the fluorescence recovery showed that, at the time scale relevant for the dithionite reduction assay, our synthesized NBD-PEG molecules are essentially membrane-impermeable (Supplementary Fig. 14b). The same batch of vesicles was then incubated with DNA scramblases, again with NBD-PEG molecules present inside and outside the vesicles. A dithionite reduction assay was carried out following the same protocol as in our lipid scrambling assays (Supplementary Fig. 15a, b). Monitoring NBD-PEG fluorescence showed that the fluorescence outside the vesicles was rapidly decreased whereas the fluorescence inside the vesicles remained almost constant over the 45-min time scale of the measurement (see Supplementary Fig. 15b–d). These experiments show that dithionite permeation through the DNA-induced pores is too slow to explain the rapid decrease in fluorescence observed in our 2C DNA scramblase experiments.

The above experimental results establish that our DNA nanostructure acts as a lipid scramblase in biological membranes at physiological pH values in vitro and that an alternative design, not capable of membrane insertion, does not produce lipid scrambling. Traces for both 1C and 2C structures shown in Fig. 4c are well described by a single exponential decay (Supplementary Fig. 16a, b), which is in agreement with a previous characterization of fully activated biological scramblases that found the dithionite reduction of the NBD dye to be the rate limiting factor in the case of rapid scrambling[27]. Approximating the number of lipids per vesicle, the overall scrambling rates per vesicle were calculated to be between $\sim(0.7–3.7) \times 10^7\,s^{-1}$ (Supplementary Fig. 16d), which is in very good agreement with the simulated scrambling rates of individual DNA scramblases.

**DNA scramblase experiments on human cancer cells**. To show the potential for in vivo applications, we tested our DNA scramblase in human cells. We incubated breast cancer cells (MDA-MB-231) for 1 h with our DNA scramblases and subsequently stained the cells with FITC-labeled annexin V, which has a high binding affinity for phosphatidylserine (PS) lipids. As the employed cells naturally possess a low level of PS in the outer membrane leaflet[31], annexin V binding to untreated cells is very low (Fig. 4e, f). Successful scrambling by our 2C DNA nanostructures would be indicated by an elevated level of surface-exposed PS resulting in increased binding of FITC-annexin V (Fig. 4g). Confocal microscopy images presented in Fig. 4h show not only that the DNA nanostructures attached to the cells but also that annexin V binding increased in their vicinity. A positive control using the apoptosis-inducing microbial alkaloid staurosporine showed a

similar maximum intensity of annexin V binding (Supplementary Fig. 17b). These results demonstrate that our DNA scramblase is able to induce lipid scrambling in human cells.

## Discussion

Experiments with pore-forming peptides determined lipid flip–flop rates between 1 and potentially $10^3$ lipids per second per peptide[32,33]. Recent in vitro experiments on TMEM16 scramblases[27], opsin[26], and rhodopsin[34] have assessed lipid flip–flop rates of $> 10^4 \, s^{-1}$ per scramblase protein under optimal conditions. However, these measured rates were limited by the dithionite-mediated NBD reduction. Recent atomistic simulations of the G protein-coupled receptor opsin determined a characteristic time scale of ~33 µs per lipid translocation event[35]. This corresponds to a possible maximum scrambling rate of $3 \times 10^4 \, s^{-1}$ in the case where dithionite is not the rate limiting factor. In contrast, we found the simulated lipid transfer rate induced by our DNA nanostructure to be in the range of $(1.9–2.6) \times 10^7 \, s^{-1}$, up to three orders of magnitude faster than reported for natural scramblases. To achieve flip–flop rates equivalent to natural scramblases the free energy barrier for lipid translocation needs to be lowered from $> 20$ kcal mol$^{-1}$ (uncatalyzed lipid transfer) to ~7 kcal mol$^{-1}$ [36]. One reason for the remarkable scrambling rates of our DNA scramblase is the reduction of the free energy barrier to approximately $1 \, k_B T$ ($\approx 0.6$ kcal mol$^{-1}$ at room temperature), one order of magnitude lower than accomplished by natural scramblases. Furthermore, the DNA-induced toroidal lipid pore is stable for much longer than transient water passages that were previously suggested to mediate spontaneous lipid flipping and flopping[37–40].

Our experimentally determined average scrambling rate of ~$1.62 \times 10^7 \, s^{-1}$ matches the simulation results very well; however, these rates can have multiple contributions. While several structures could insert and scramble lipids at the same time, they might only transiently insert and therefore not actively contribute for extended periods. Calculations estimating the mean lifetime $\langle \tau \rangle$ for a freely diffusing phospholipid to encounter a single, immobile flippase have previously been employed to gauge characteristic flipping times assuming every encountered lipid is flipped, and inter-leaflet translocation is not rate limiting[41]. Applied to our DNA scramblase embedded in a POPC vesicle with the average diameter of the vesicles used to determine scrambling rates (see Supplementary Fig. 16b), the calculated mean lifetime is 3.6 min, which is only slightly lower than the averaged experimental value of 4.8 min (for detailed calculations see Supplementary Note 5). Furthermore, the scrambling rate defined as the total number of lipids in a vesicle divided by $\langle \tau \rangle$ is calculated as $1.60 \times 10^7 \, s^{-1}$ assuming only a single DNA nanostructure is active, which agrees very well with the simulated and experimentally observed rates (Supplementary Table 5). These results, together with the calculated low energy barrier for lipid translocation, suggest, that lipid diffusion in our GUVs is not rate limiting even if only one DNA scramblase is active. If multiple structures scramble lipids simultaneously, a faster fluorescence reduction would be expected but the slow kinetics of the dithionite reduction are ultimately rate limiting.

In summary, we have shown that our synthetic DNA nanostructure can reproduce the biological function of a scramblase protein by inducing mixing of lipids that reside on opposite leaflets of a biological membrane in vitro and in human cells. Our synthetic DNA scramblase mixes lipids much more rapidly, outperforming biological scramblases by up to three and reported artificial scramblases by up to six orders of magnitude[36,42,43]. These exceptional rates are promoted by a stable DNA-induced toroidal lipid pore directly interconnecting the membrane leaflets without any substrate specificity or covalent bond formation. By equipping them with an activation mechanism and the ability to target plasma membranes of specific cell types, our DNA scramblase can be made suitable for biomedical applications with the scrambling activity being controlled by the geometry of the toroidal lipid pore. On demand, target cell-specific lipid scrambling can aid patients suffering from impaired lipid scrambling or be used to trigger phagocytic uptake of PS-exposing intruder cells by macrophages, including cells carrying cancer specific plasma membrane antigens. Furthermore, the mechanism of our DNA scramblase is independent of any cell-specific apoptosis pathways, making it applicable to a broad range of cell types. Control over lipid homeostasis by synthetic DNA nanostructures opens up a yet to be explored direction for designing personalized drugs and therapeutics for a variety of health conditions. Ultimately, the ability to outperform naturally evolved proteins allowed for a glimpse at the tremendous opportunities still to be explored in nucleic acid-based nanotechnology.

## Methods

**All-atom MD simulations.** The caDNAno design of the DNA nanostructure (Supplementary Fig. 1a) was converted to an idealized all-atom representation and embedded into a lipid bilayer membrane using a previously described method[24]. The resulting model was merged with a rectangular volume of electrolyte solution, minimized, and equilibrated in the constant number of particles, temperature and pressure ensemble. The simulations employed the CHARMM36 parameter set[44] and were carried out using both NAMD[45] and Anton[46]. Supplementary Note 1 provides a detailed account of the simulation procedures.

**BD simulations.** The BD simulations were performed using our in-house GPU-accelerated program Atomic Resolution Brownian Dynamics[47]. In our BD simulation, lipid head groups were modeled as point particles that interacted with each other via a repulsive potential. All other components of the systems, including the DNA nanostructure, the lipid tails, and the electrolyte solution, were modeled implicitly. A position-dependent potential was used to confine motion of the lipid head groups to the volume they occupied in the all-atom simulations; the all-atom MD trajectories were also used to determine position-dependent diffusivity of the head groups. In Supplementary Notes 2 and 3, we provide a detailed description of the simulation procedures.

**DNA nanostructure assembly.** All reagents were acquired from Sigma-Aldrich if not stated otherwise. DNA nanostructures were designed using caDNAno[48] and sequences optimized to minimize undesired hybridization sites. All DNA oligo-nucleotides were acquired from Integrated DNA Technologies (IDT). Unmodified DNA strands (purified by standard desalting) and 3′-Cy3-modified strands (HPLC-purified) were ordered pre-diluted to 100 µM in IDTE buffer (10 mM Tris, pH 8.0, 0.1 mM EDTA) and stored at −20 °C. Cholesterol-tagged DNA strands were modified at the 3′-end via a 15 atom triethyleneglycol spacer, purchased HPLC-purified, diluted to 100 µM in Milli-Q water (Merck Millipore) upon arrival and stored at 4 °C. DNA nanostructures were assembled analogously to a previously described protocol[16]. Briefly, an equimolar mixture of eight DNA strands was prepared at 1 µM final concentration per oligonucleotide in TE20 buffer (10 mM Tris, 1 mM EDTA, 20 mM MgCl₂, pH 8.0). If desired, cholesterol- or Cy3-modified strands were introduced by omitting the equivalent unmodified oligo-nucleotide and adding the modified one into the assembly mix instead. For cholesterol-modified DNA strands, stock solutions were heated to 55 °C for 10 min prior to addition to the assembly mix. Folding of DNA nanostructures was performed by heating the oligonucleotide mixture to 85 °C to ensure complete strand separation, and subsequent cooling to 25 °C via an 18 h temperature gradient using a ProFlex™ PCR thermal cycler (Thermo Fisher Scientific). Folded structures were stored at 4 °C protected from light.

**PAGE of DNA nanostructures.** The gel was cast at a concentration of 10% polyacrylamide supplemented with 0.5 × Tris-borate-EDTA (TBE) and 11 mM MgCl₂. Per 15 ml gel mixture, 150 µl of 10% ammonium persulfate solution and 10 µl N,N,N′,N′′-tetramethylethylenediamine were added to initiate polymerization. 2 µl of DNA nanostructures at 1 µM were mixed with 0.4 µl custom-made 6 × loading dye (6 ×: 15% Ficoll 400, 0.9% Orange G diluted in TE20 buffer) and 2 µl of the mixture were loaded into the well. The gel was run in a Mini-PROTEAN Tetra Cell (Bio-Rad) for 90 min at 100 V in 0.5 × TBE supplemented with 11 mM MgCl₂ and afterwards stained using GelRed (Biotium) and the bands visualized via UV-transillumination. The gray scale of the acquired image was inverted and subsequently the background subtracted using the rolling ball method (radius = 300 pixel) in Fiji.

**Preparation of lipid vesicles**. GUVs were prepared by electroformation using a Nanion Vesicle Prep Pro setup. 1-palmitoyl-2-oleoyl-sn-glycero-3-phosphocholine lipid (POPC; Sigma-Aldrich) and 1-palmitoyl-2-{6-[(7-nitro-2-1,3-benzoxadiazol-4-yl) amino] hexanoyl}-sn-glycero-3-phosphocholine (NBD-PC; Avanti Polar Lipids) were dissolved in chloroform and mixed in a w/w ratio of 200:1 (POPC: NBD-PC). 100 µl of the lipid mixture at 5 mg ml$^{-1}$ was spin-coated on the conducting surface of an indium tin oxide (ITO)-coated glass slide (Nanion/Vision-Tek). Chloroform was evaporated for 1 h in a desiccator following which 600 µl of sucrose buffer (100 mM sucrose, 20 mM HEPES at pH 7.4) was deposited within the O-ring chamber which was subsequently sealed with another ITO-coated slide (conducting surface facing the other). The electroformation chamber was then connected to the Nanion Vesicle Prep Pro and the electroformation protocol proceeded in three steps: (i) The A/C voltage increased linearly from 0 to 3.2 V peak-to-peak (p-p) at 10 Hz over 1 h, (ii) the voltage stayed at 3.2 V p-p and 10 Hz for 50 min, (iii) the frequency decreased linearly to 4 Hz over 10 min and was maintained for another 20 min. Electroformation was carried out at 37 °C and vesicles were stored at 4 °C protected from light. Vesicles were not used longer than 36 h after formation.

**Dithionite reduction assay**. Assembled DNA nanostructures (1 µM) with either one or two cholesterol modifications were mixed with 0.5% poly(ethylene glycol) octyl ether (OPOE), pre-diluted in TE20, in a 7:1 ratio and incubated for 2 min at room temperature. The mixture was then diluted in glucose buffer (100 mM glucose, 4 mM MgCl$_2$, 20 mM HEPES titrated to pH 7.4 with KOH) and added to 20 µl GUV solution at a final concentration of 100 nM DNA nanostructures. Samples were then incubated for 90–120 min on a 1% BSA-coated glass coverslip within an incubation chamber (Grace Bio-Labs) at room temperature allowing the vesicles to settle to the bottom due to the density gradient between the intravesicular sucrose and extravesicular glucose as well as the cholesterol-modified DNA nanostructures to anchor into the lipid membrane. Dithionite was dissolved in 1 M Tris at pH 10 at a concentration of 1 M and then pre-diluted in 50 mM glucose, 4 mM MgCl$_2$, and 20 mM HEPES pH 7.4 to a concentration of 15 mM dithionite freshly before each experiment. To initiate NBD dye reduction, 30 µl of diluted dithionite solution were carefully added to the incubated vesicles to a final concentration of 4.5 mM dithionite at approximately 1 min after starting the recording. Chambers were covered throughout with a glass slide to prevent evaporation except when the dithionite solution was added. At all times in the protocol at least 4 mM MgCl$_2$ were present to keep the DNA nanostructures stable over time (see Supplementary Table 3 for detailed buffer conditions). Images were acquired on an Olympus FluoView filter-based FV1200F-IX83 laser scanning microscope using a 60 × oil immersion objective (UPLSAPO60XO/1.35). NBD excitation was performed using a 25 mW 473 nm laser diode at 1% laser power and emission was collected between 490 and 525 nm. Cy3 was excited with a 1.5 mW 543 nm HeNe laser at 5% laser power and emitted light collected between 560 and 660 nm. For statistical analysis a z-stack (slice thickness 300 nm) was recorded of the field of view before and 35 min after dithionite addition with separate excitation of the 473 and 543 nm laser lines at a sampling speed of 2.0 µs pixel$^{-1}$. For single vesicle fluorescence reduction traces, vesicles of similar size were kept in focus and images were recorded every 10 s in between the z-stacks while exciting with both lasers simultaneously. Images were analyzed using Fiji. Vesicles were identified and located from the fluorescence signal collected from Cy3-labeled DNA nanostructures by applying a ring-shaped selection area over the fluorescent ring at a height close to the equatorial plane. NBD fluorescence intensity for each vesicle was then determined by measuring the mean gray value of the equivalent area in the respective images of the NBD emission channel. Values were background subtracted by measuring and averaging over three areas without vesicles. Intensities per vesicle were normalized to the average intensity of the first five data points of each trace.

**Annexin V staining experiments on human cells**. MDA-MB-231 cells were acquired from the Cancer Research UK Cambridge Institute Biorepository where the cells were authenticated by multiplex PCR and short tandem repeat (STR) profiling including detection of mouse cell contamination. Cells were maintained in Dulbecco's Modified Eagle's Medium (DMEM; Sigma-Aldrich) supplemented with 10% (v/v) heat-inactivated fetal calf serum (FCS; Thermo Scientific) at 37 °C and 5% CO$_2$. A concentration of 30,000 cells per 250 µl medium was seeded on a cover glass placed in a well of a 48-well plate (day 0) and grown for two nights under the same conditions as stated above. Afterwards, cells were washed once with phosphate buffer saline (PBS) and then covered again in fresh medium. DNA nanostructures with two cholesterols and two Cy3 tags were assembled as described above but in PBS at pH 7.4 supplemented with 8 mM MgCl$_2$ instead of TE20 buffer. 120 µl of assembled structures were added to cells (prepared as described above) in the well plate (final structure concentration 324 nM) and incubated for 1 h at 37 °C and 5% CO$_2$. For the negative control performed in parallel, only the employed DNA folding buffer without DNA nanostructures was added. Subsequently, cells were washed with 500 µl of 1 × annexin V binding buffer (Abcam) and then incubated with 500 µl of 1 × FITC-labeled annexin V (Abcam) in binding buffer for 5 min (in accordance with the protocol provided by the manufacturer). After staining, cells were washed with 500 µl binding buffer once again and then

fixed in 250 µl of 4% formaldehyde in binding buffer for 15 min on ice, followed by three washing steps with 250 µl binding buffer before being stored in the fridge overnight. On day four the cover glasses were transferred onto microscope slides by mounting them with Mowiol[49]. For this, 6 g of glycerol, 2.4 g of Mowiol powder (Calbiochem) and 6 ml of distilled water were added to 12 ml of 0.2 M Tris buffer (pH 8.0) and stirred for 4 h. The solution was then left to rest for an additional 2 h. Subsequently, the mixture was incubated for 10 min in a 50 °C water bath and finally centrifuged for 15 min at 5000 × g. After removing the supernatant, the solution was stored at −20 °C before usage. Images were acquired on the same confocal microscope as described for the dithionite reduction assay except that a 20 × air objective was used (UPLSAPO20X/0.75). Filter set and laser power for Cy3-labeled DNA nanostructures were kept the same and parameters used for the NBD dye were applied for imaging FITC-labeled annexin V as well. Detector voltages for both channels were kept fixed and were the same for all experiments. Z-stacks (slice thickness 500 nm) of cells were acquired with separate excitation of the 473 and 543 nm laser lines (sampling speed 2.0 µs pixel$^{-1}$). The bright field images were obtained by acquisition of the transmitted light of the 543 nm laser. Analysis was performed using Fiji.

**Data availability**. The data that support the findings of this study are available from the corresponding authors upon reasonable request.

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

## Acknowledgements

The authors thank Dr. Cally Haynes for critical reading of the manuscript and Dr. Karolis Misiunas for insightful discussions regarding scrambling rate calculations. A.A., C.M., J.Y., and C.Y.L. acknowledge the support from the National Science Foundation under Grants DMR-1507985, PHY-1430124, and EEC-1227034, National Institutes of Health grant P41-GM104601 and the supercomputer time provided through XSEDE Allocation Grant MCA05S028 and the Blue Waters petascale supercomputer system (UIUC). Anton 2 computer time was provided by the Pittsburgh Supercomputing Center through Grant R01-GM116961 from the National Institutes of Health. The Anton 2 machine at PSC was generously made available by D.E. Shaw Research. U.F.K. and K.A.N. were supported by an ERC consolidator grant (DesignerPores 647144). K.N.B. acknowledges the DAAD for a PROMOS scholarship. K.G. was supported by the Winton Programme for the Physics of Sustainability, Gates Cambridge, and the Oppenheimer Ph.D. studentship. A.O. acknowledges funding from the EPSRC and from the Vice-Chancellor's Award from the Cambridge Trust.

## Author contributions

A.O. and C.Y.L. contributed equally to this work. C.Y.L. performed all-atom and BD simulations and analysis. J.Y. carried out all-atom MD simulations and assisted with their analysis. C.M. wrote the BD simulation package, helped setup the BD simulation and analyzed all-atom MD trajectories. A.O. designed, performed and analyzed DNA nanostructure, lipid vesicle and cell experiments. K.A.N. prepared lipid vesicles and advised on handling. K.N.B. carried out cell culture and preparation. K.G. provided vital input for the design and validity of experiments. U.F.K. and A.A. supervised and advised throughout the work. C.Y.L. wrote the initial draft, A.O. added the experimental part and A.O., U.F.K., and A.A. finalized the manuscript.

## Additional information

**Competing interests:** The authors declare no competing interests.

