## [Peer Review File · Nature Communications]

Reviewers' comments:

Reviewer #1 (Remarks to the Author):

This paper represents a landmark study. The experiments and simulations seem to be carefully performed using appropriate techniques.

I commend the authors on a great piece of work.

Reviewer #2 (Remarks to the Author):

This is a well-written paper describing an innovative series of experiments/simulations.

While the simulations and cell-based assays clearly indicate the ability of the synthetic DNA-based nanostructure to scramble lipids, the key GUV experiments lack an important control. These experiments make use of the standard dithionite reduction protocol to probe the accessibility of fluorescent NBD-labeled phospholipids on the external leaflet of membranes. However, in the presence of the DNA nanopore, dithionite would be expected to permeate into the vesicle interior accessing lipids on the inner face. Thus, the GUV experiments do not provide evidence for lipid scrambling – the authors must remedy this issue before the paper can be considered in further detail.

Additional points:

1. The authors make a great deal that their DNA nanostructure provides record scrambling rates. Before this statement can be made, they should carefully look into the reports of lipid scrambling by peptides, some of which are presumed to form toroidal pores in the membrane. This literature is not cited or discussed in the paper and must be included. For example:

Matsuzaki K, Murase O, Fujii N, Miyajima K. (1996) *Biochemistry* 35:11361-8.

Fattal, E., Nir, S., Parente, R.A., Szoka, F.C. (1994) *Biochemistry* 33:6721-31.

2. On lines 179-180 the authors comment that lipid diffusion towards the pore does not limit the rate of scrambling. An explanation of this result is that the lateral and transverse diffusion coefficients are the same, as might be expected if the pore simply fused the two leaflets of the bilayer, around a bend as depicted in Fig. 4E. An article by Gummadi and Menon (2002) *J. Biol. Chem.* 277:25337-43 presents a calculation of the rate of transport in this context (the calculation is in a footnote in the discussion of the Gummadi paper) and it would be appropriate to cite this paper and discuss the calculation.

3. On line 270, the authors quote published data on the rates of lipid scrambling by TMEM16 proteins and opsin. This rate should be amended to >10,000 per second (it cannot be measured except as a lower limit, on account of the slow rate of dithionite-mediated reduction of the NBD tracer lipid). The conclusion on line 272 should therefore be modified to say 'at most 3 orders of magnitude' or something to that effect. A similar correction should be made on line 295.

4. The simulations are done in DPhPE or DPhPC lipid bilayers. Please explain why this system was chosen. In contrast, the GUV experiments are done with POPC membranes.

5. The simulations are done under quite unnatural salt conditions – 1 M KCl. Please explain.

Minor points:

1. It would be appropriate to cite the recent review by Pomorski and Menon (*Prog. Lip. Res.* 64:69-84 (2016)) as it provides a good assessment of the kinetics of scrambling by natural

scramblases.

2. Line 299: PS exposure does not trigger apoptosis, rather it triggers the phagocytic uptake of PS-exposing cells by macrophages.

3. Line 383: dithionite reduces or bleaches the NBD dye; it does not quench the fluorescence, rather it eliminates the fluorescence.

Reviewer #3 (Remarks to the Author):

In this interesting article, the authors report on the increased transport of lipids from one leaflet of a lipid bilayer to another, which is induced by the presence of an artificial DNA membrane nanostructure. The authors had previously investigated membrane incorporation of such and similar structures as artificial nanopores. In the present work they compare the action of the structures with naturally occurring enzymes – scramblases – which have the same function of increasing lipid flipping in the bilayer. This highlights a completely novel aspect of membrane-spanning DNA nanostructures, which may also have biomedical implications. The paper contains an interesting mix of computational and experimental studies of the process.

In a revised version of the manuscript the following points should be addressed:

- is the nanostructure used for this work in any way different to previously published DNA pores? The text gives the possibly wrong impression that the structure was specifically “designed” for scramblase action.

- in what way are the results dependent on the chosen lipids and also the lipid phase? Conceivably, other lipid mixtures under different conditions, possibly phase-separated, would have a much higher background flip-flop activity. For instance, does the signal for the control in Fig. 4C drop below 0.5 on the long term because of normal flip-flop events?

- maybe I missed this point, but when comparing the scramblase efficiency – is it normalized to transferred lipids per pore? per area? Do you really compare the efficiency of a single scramblase with a single DNA nanostructure, or is this an effect of different densities, etc.?

- the authors claim (on page 13) that scramblase action by the DNA nanostructure is better because of the larger diameter toroidal lipid pore. This should also be true for other DNA pore structures by the same group and by others. Is there any (e.g., computational) evidence for general scramblase action of all of these structures? Or is the four-helix structure special?

- in Fig. 4F it would be good to see the results of the negative control experiment also in the main paper.

- the title is really a little too catchy (but this is personal taste) – it sounds a little like a newspaper headline.

- it is interesting pointing out that this specific “reaction” of lipid flipping is “catalyzed” much better than by scramblases. Nevertheless, this is a very special reaction – no substrate specificity, no covalent bond formation involved, etc. – and probably the hopes for creating artificial enzymes with similar efficiency for other reactions are rather low. It would therefore be good to differentiate this point a little better within the text.

Reviewer #4 (Remarks to the Author):

Ohmann et al. have developed a synthetic molecular DNA-based complex able to foster the translocation of lipids across lipid membrane and cell membrane structures. Using both computer simulations and experiments they demonstrate the proof of principle that these synthetic molecules can increase the translocation rate more than any known biological (scramblase) protein. The findings are unique and the concept used is novel. Given that lipid translocation is in a central position in many biological processes such as apoptosis, the results discussed in the paper can be considered to be important, and of broad interest.

Yet, there are several points that the paper does not discuss to a sufficient degree.

While the findings are appealing, the paper does not provide a convincing explanation as to what exactly happens, and why?

The paper shows that the DNA nanostructure explored in the study increases the lipid translocation rate very significantly. However, what exactly happens, how does the translocation process take place? What is the reaction/transition path of lipids during the translocation process? How do the lipids orient themselves during the translocation process, such as, do they for instance expose their polar head group to the DNA nanostructure in a manner that would show electrostatic interactions to predominate the translocation process? Do the lipids translocate one by one, or are there concerted dynamic motions involved, leading to translocation of clusters of lipids in a concerted fashion?

Moving on, the question why? What are the cause and the driving force for the observed motion? The authors could have used their simulation data for the observed translocation processes along the DNA nanostructure as a basis for free energy profile calculations, and by doing this, they could have computed the free energy profile for translocation across a membrane and determined how it depends on lipid-DNA interactions (in varying positions along the DNA nanostructure surface)? What is the free energy barrier for translocation, where is the largest barrier along the transition path, and what causes it? If the translocation is considered at several temperatures, is it the enthalpic or the entropic component that dominates the translocation process and the free energy barrier that apparently is much lower than the barrier associated with lipid translocation in the absence of the DNA nanostructure?

Unfortunately these two questions (and the many follow-up questions resulting from these two key questions) are not addressed adequately. The paper demonstrates a unique finding, which is valuable, but the paper is not yet able to describe what really happens and why?

As to more minor questions, the authors are encouraged to consider the following.

The paper shows that in order to be functional, the DNA nanostructure should have two cholesterol tags attached to it. Why? Why one is not sufficient? Again calculations of free energy profiles associated with translocation paths could provide a means to explain the matter.

If the reviewer is not mistaken, the reported translocation rates describe absolute numbers across a membrane (from upper to lower, and from lower to upper leaflet). However, what is the net rate across the membrane? Is it different from zero?

While this may be speculative, is there reason to assume, based on the results reported in the paper, that the DNA-mediated translocation rates would be lipid dependent?

Response to reviewers regarding:

**A synthetic enzyme built from DNA flips 10^7 lipids per second
in biological membranes**

NCOMMS-17-27368-A

Alexander Ohmann, Chen-Yu Li, Christopher Maffeo, Kareem Al Nahas, Kevin N. Baumann, Kerstin
Göpfrich, Jejoong Yoo, Ulrich F. Keyser, Aleksei Aksimentiev

Reviewer #1 (Remarks to the Author):

This paper represents a landmark study. The experiments and simulations seem to be carefully performed using appropriate techniques.

I commend the authors on a great piece of work.

Response: We thank the reviewer for his or her appreciation of our work.

Reviewer #2 (Remarks to the Author):

This is a well-written paper describing an innovative series of experiments/simulations.

While the simulations and cell-based assays clearly indicate the ability of the synthetic DNA-based nanostructure to scramble lipids, the key GUV experiments lack an important control. These experiments make use of the standard dithionite reduction protocol to probe the accessibility of fluorescent NBD-labeled phospholipids on the external leaflet of membranes. However, in the presence of the DNA nanopore, dithionite would be expected to permeate into the vesicle interior accessing lipids on the inner face. Thus, the GUV experiments do not provide evidence for lipid scrambling – the authors must remedy this issue before the paper can be considered in further detail.

Response:

We thank the reviewer for pointing out the importance of this necessary control to ascertain that the observed fluorescence reduction in the GUV experiments is not primarily facilitated by dithionite permeating into the vesicle interior via the DNA nanostructure. We now reanalyzed our MD simulations and performed new control experiments that show that dithionite transport through the DNA nanostructures is much slower and we indeed see lipid scrambling.

Analysis of MD simulations:

As our nanostructure is made from DNA, it is highly negatively charged throughout. Therefore, negative ions are passing less frequently than positive ions through a toroidal pore because of the electrostatic repulsion from the DNA nanostructure. We have previously reported on the significantly reduced permeation of Cl⁻ ions compared to K⁺ ions on a larger, membrane-inserted DNA nanostructure [1]. Analysis of our all-atom MD simulations of the DNA scramblase design showed a 93 % reduction of spontaneous Cl⁻ ion permeation through the DNA nanopore in comparison to K⁺ ion permeation, confirming the previously reported ion selectivity of membrane-inserted DNA nanostructures. In the dithionite reduction assay, the NBD-reducing dithionite anion [S₂O₄]²⁻ is larger than a Cl⁻ ion and, most importantly, is twice negatively charged. Because a dithionite ion has a greater negative charge and a larger volume, and because the buffer solution used in experiment is of a low ionic strength (4 mM MgCl₂), the charge of the DNA nanostructure is expected to present a large energy barrier to dithionite permeation through the toroidal pore.

Inspired by the literature where it was shown that dithionite does not leak through natural scramblase proteins [2], we performed new experiments that confirm the slow diffusion of dithionite through the DNA-induced nanopores. Our experiments show that dithionite diffusion through such lipid nanopores is too slow to explain the GUV assays we present. The new control measurements are included in the revised manuscript as Supplementary Figs. 14 and 15.

New experiments:

In order to assess dithionite permeation, we created an NBD-dye conjugate that could NOT pass through our DNA scramblase. To this end, we synthesized a test molecule from NBD and a 24-

unit polyethylene glycol (NBD-PEG, Supplementary Fig. 14a). The covalent attachment of the PEG chain increases the hydrodynamic radius and reduces hydrophobicity of the NBD dye, preventing direct permeation of the NBD-PEG construct through the membrane and through the toroidal pores produced by our DNA scramblase. We experimentally characterized the permeation of the NBD-PEG construct by forming POPC vesicles having NBD-PEG encapsulated inside. After incubation of these vesicles with 2C DNA nanostructures (our DNA scramblase design), the NBD dyes inside the vesicles were photobleached. Measurements of the fluorescence recovery have shown that, at the time scale relevant for the dithionite reduction assay, our synthesized NBD-PEG molecules are essentially membrane-impermeable (Supplementary Fig. 14b). The same batch of vesicles was then incubated with 2C DNA nanostructures and having the NBD-PEG molecule present inside and outside of the vesicles. Next, a dithionite reduction assay was carried out following the same protocols as in our lipid scrambling assays while we monitored the NBD fluorescence inside and outside the vesicles (Supplementary Fig. 15a,b). While the fluorescence outside the vesicles was observed to rapidly decrease, the fluorescence inside the vesicles remained almost constant over the 45-minute time scale of the measurement (see Supplementary Fig. 15b-d). A slight decrease of intravesicular fluorescence is interpreted to be a combined effect of (1) the synthesized NBD-PEG molecules not being completely membrane-impermeable (see Supplementary Fig. 14b), (2) the decreased (by dithionite reduction) background intensity outside the vesicles also affecting the detected signal inside the vesicles, and (3) a contribution of dithionite diffusing through the DNA-induced pore. However, the difference in the rates of fluorescent reduction inside and outside of the vesicles provides strong evidence that dithionite diffusion through the DNA nanostructures is negligible. As dithionite does not cross membranes in significant quantities, the reason for the fluorescence reduction observed in our assays including our 2C structures is rapid lipid scrambling activity. In other words, our control experiments have shown that dithionite permeation through the DNA-induced pores is too slow to explain the rapid decrease in fluorescence observed in the 2C DNA scramblase experiments.

We have incorporated a paragraph summarizing the above data and discussion on pages 15 and 16 in the main manuscript and added two experimental figures to the supplementary information (Supplementary Figs. 14 and 15).

Additional points:

1. The authors make a great deal that their DNA nanostructure provides record scrambling rates. Before this statement can be made, they should carefully look into the reports of lipid scrambling by peptides, some of which are presumed to form toroidal pores in the membrane. This literature is not cited or discussed in the paper and must be included. For example:

Fattal, E., Nir, S., Parente, R.A., Szoka, F.C. (1994) *Biochemistry* 33:6721-31.
Matsuzaki K, Murase O, Fujii N, Miyajima K. (1996) *Biochemistry* 35:11361-8.

Response: We value the suggestion of the reviewer to expand the extent of our literature review to include previously reported high scrambling rates. In Fattal *et al.* the highest reported scrambling rate was for the synthetic peptide GALA at $k \geq 1 \text{ s}^{-1}$ (Table 2 in their publication). In Matsuzaki *et al.*, large unilamellar vesicles with an average diameter of $\sim 90 \text{ nm}$ symmetrically labeled with NBD-PE were fully reduced in their fluorescence by dithionite within $\sim 1.5 \text{ min}$ in the presence of the pore-forming peptide Magainin 2. Such vesicles are composed of approximately 70,000 lipids which would equate to an overall scrambling rate of $\sim 800 \text{ lipids/s}$. However, the peptide was added at quite high concentrations of $3 \mu\text{M}$, suggesting that the scrambling rate per peptide is possibly lower.

We have included both references together with a remark summarizing the above analysis in the main manuscript on page 17. The paragraph furthermore discusses recently published work reporting higher scrambling rates, including a most recent atomistic simulation analysis of a natural scramblase [3]. We have furthermore included the suggested review by Pomorski and Menon providing a recent overview of the literature regarding scrambling kinetics (page 18 top and page 19 bottom).

The highest reported absolute rate at $3 \times 10^4 \text{ s}^{-1}$ in a case where dithionite was not rate limiting could be inferred from the above-mentioned MD simulation analysis [3]. This rate still falls short of our reported rate by approximately three orders of magnitude. However, we see the reviewer's concern regarding the claim of record rates and therefore changed the title: "A synthetic enzyme built from DNA flips 10^7 lipids per second in biological membranes".

*2. On lines 179-180 the authors comment that lipid diffusion towards the pore does not limit the rate of scrambling. An explanation of this result is that the lateral and transverse diffusion coefficients are the same, as might be expected if the pore simply fused the two leaflets of the bilayer, around a bend as depicted in Fig. 4E. An article by Gummadi and Menon (2002) *J. Biol. Chem.* 277:25337-43 presents a calculation of the rate of transport in this context (the calculation is in a footnote in the discussion of the Gummadi paper) and it would be appropriate to cite this paper and discuss the calculation.*

Response: We thank the reviewer for suggesting the inclusion of the mentioned calculation. We have added an extensive section detailing the necessary steps and calculations using this theoretical model to our case in the Supplementary Material on page 22. We find that the

scrambling rate is agreeing extremely well with the model. We believe that the helpful suggestion by the reviewer makes the discussion on the rates even stronger.

In response to the reviewer's comment, we added a paragraph to the discussion, interpreting the obtained results which has allowed us to further support our experimentally determined rates with this theoretical model and infer that even in the larger experimental systems, lipid diffusion is not rate limiting. Within the context of this discussion, we have referenced the article by Gummadi and Menon in the main manuscript (please see page 18) which further strengthens the message of our work.

3. On line 270, the authors quote published data on the rates of lipid scrambling by TMEM16 proteins and opsin. This rate should be amended to >10,000 per second (it cannot be measured except as a lower limit, on account of the slow rate of dithionite-mediated reduction of the NBD tracer lipid). The conclusion on line 272 should therefore be modified to say 'at most 3 orders of magnitude' or something to that effect. A similar correction should be made on line 295.

Response: The reviewer is absolutely correct in his or her comment on the previously reported scrambling rates. We therefore amended our manuscript to correctly incorporate previously reported results and adequately compare them to our findings. Please find the corrected text on pages 17 and 19.

4. The simulations are done in DPhPE or DPhPC lipid bilayers. Please explain why this system was chosen. In contrast, the GUV experiments are done with POPC membranes.

Response: The DPhPC lipid bilayer was chosen for our MD simulations to match previous experimental studies that investigated the use of DNA nanostructures as transmembrane ion channels [4,5]. Specifically, we performed MD simulations to examine ion conductance of a variety of DNA nanostructures, but, unexpectedly, discovered the lipid scrambling phenomenon. To check how lipid scrambling depends on the type of lipid molecules, we carried out another simulation using a DPhPE bilayer, finding lipid scrambling to occur about 3 times faster. The experimental part of the study was developed after the simulations were completed. In the GUV experiments, POPC lipids were employed for two reasons. In contrast to the simulations the experiments required NBD-PC lipids as fluorescent tracers. Therefore, forming the vesicles from POPC lipids provided a better consistency in the lipid tail chemistry between the vesicle-forming lipids and the tracer lipids making the tracer lipids a more accurate representation of the bulk lipids. Secondly, previous scrambling experiments employed vesicles formed from egg PC [2,6] which is a mixture of lipids with the highest content being 16:0 and 18:1 PC tails [7] which are exactly the fatty acid tail lengths of POPC. Additionally, these fatty acid tails are the most abundant in naturally occurring phospholipid mixtures [8]. Therefore, both simulations and experiments allow for best comparability to previously published and peer-reviewed results. However, given a fairly general character of lipid scrambling facilitated by the formation of a DNA-induced toroidal pore, we expect that differences in the lipid tail structure to have effects on the rates well below factors of 10. We included a brief discussion on the advantage of using POPC lipids in the experiments in a paragraph on page 13.

5. *The simulations are done under quite unnatural salt conditions – 1 M KCl. Please explain.*

Response: Again, this specific ion condition was chosen to match previous experimental studies that investigated the use of DNA nanostructures as transmembrane ion channels [4,5]. The specific experimental assay (which was developed after the simulations were performed) required the use of electrolyte of a lower ion concentration in order to employ electroformation for preparation of giant unilamellar vesicles which does not allow for the presence of high concentrations of ions. Because lipid scrambling is facilitated by the formation of the toroidal nanopore, ionic concentration will have only minor effects on scrambling of electrically neutral lipids as long as the DNA nanostructure remains structurally stable.

Minor points:

1. *It would be appropriate to cite the recent review by Pomorski and Menon (Prog. Lip. Res. 64:69-84 (2016)) as it provides a good assessment of the kinetics of scrambling by natural scramblases.*

Response: We thank the reviewer for pointing out this comprehensive recent review. We included this reference on page 18 (top) for the comparison regarding the free energy barrier between our DNA nanostructure and natural scramblases, and on page 19 (bottom) for the comparison to scrambling rates in the literature.

2. *Line 299: PS exposure does not trigger apoptosis, rather it triggers the phagocytic uptake of PS-exposing cells by macrophages.*

Response: We have amended the sentence as per suggestion by the reviewer and incorporated the correct biological consequence of PS-exposure.

3. *Line 383: dithionite reduces or bleaches the NBD dye; it does not quench the fluorescence, rather it eliminates the fluorescence.*

Response: We thank the reviewer for pointing out the inaccurate formulation used in the manuscript which could have confused the reader. We remedied this by changing the formulation to “NBD dye reduction” or “fluorescence reduction” throughout the manuscript as well as the supplementary material.

References:

1. Yoo, J. & Aksimentiev, A. Molecular Dynamics of Membrane-Spanning DNA Channels: Conductance Mechanism, Electro-Osmotic Transport, and Mechanical Gating. *J. Phys. Chem. Lett.* **6**, 4680–4687 (2015).
2. Malvezzi, M. et al. Ca²⁺-dependent phospholipid scrambling by a reconstituted TMEM16 ion channel. *Nat. Commun.* **4**, 2367 (2013).
3. Morra, G. et al. Mechanisms of Lipid Scrambling by the G Protein-Coupled Receptor Opsin. *Structure* **26**, 356–367.e3 (2018).

4. Göpfrich, K. et al. Ion Channels Made from a Single Membrane-Spanning DNA Duplex. *Nano Lett.* **16**, 4665–4669 (2016).
5. Göpfrich, K. et al. Large-Conductance Transmembrane Porin Made from DNA Origami. *ACS Nano* **10**, 8207–8214 (2016).
6. Menon, I. et al. Opsin is a phospholipid flippase. *Curr. Biol.* **21**, 149–153 (2011).
7. Egg PC (840051). *Avanti Polar Lipids* Available at: <https://avantilipids.com/product/840051/>. (Accessed: 28th March 2018)
8. Nakano, M. *et al.* Flip-flop of phospholipids in vesicles: kinetic analysis with time-resolved small-angle neutron scattering. *J. Phys. Chem. B* **113**, 6745–6748 (2009).

Reviewer #3 (Remarks to the Author):

In this interesting article, the authors report on the increased transport of lipids from one leaflet of a lipid bilayer to another, which is induced by the presence of an artificial DNA membrane nanostructure. The authors had previously investigated membrane incorporation of such and similar structures as artificial nanopores. In the present work they compare the action of the structures with naturally occurring enzymes – scramblases – which have the same function of increasing lipid flipping in the bilayer. This highlights a completely novel aspect of membrane-spanning DNA nanostructures, which may also have biomedical implications. The paper contains an interesting mix of computational and experimental studies of the process.

Response: We thank the reviewer for finding our work novel and having potential biomedical implications.

In a revised version of the manuscript the following points should be addressed:

Question 1: *is the nanostructure used for this work in any way different to previously published DNA pores? The text gives the possibly wrong impression that the structure was specifically “designed” for scramblase action.*

Response 1: The nanostructure used in our work has not been previously described in the literature. Although there could be many other DNA nanostructures that could catalyze lipid scrambling by means of toroidal pore formation, we chose our four-helix design for our experimental and simulation studies because the design lacks a central pore (which would compromise our dithionite assay) but still has a large enough circumference to produce detectable lipid scrambling activity. On page 8 (top) of the revised manuscript, we now express our expectation that other lipid-spanning nanopores should exhibit lipid-scrambling activity if their embedment is realized by means of local hydrophobic anchors.

Question 2: *in what way are the results dependent on the chosen lipids and also the lipid phase? Conceivably, other lipid mixtures under different conditions, possibly phase-separated, would have a much higher background flip-flop activity. For instance, does the signal for the control in Fig. 4C drop below 0.5 on the long term because of normal flip-flop events?*

Response 2: We thank the reviewer for raising this intriguing question. Previous experiments of lipid flipping have shown that the time scale of spontaneous lipid flip-flop indeed does depend on the employed lipid type with half-times typically being on the order of hours to days [1]. Advantageously for our performed experiments, POPC has been shown to have a half-time of spontaneous lipid flip-flop of > 1000 h in reconstituted vesicles [2]. The latter publication furthermore discusses that for DMPC lipids, moving from the lipid disordered to the lipid ordered phase, by incorporating 40 % cholesterol into the membrane, almost completely eradicates spontaneous lipid flip-flop due to the increased rigidity of the membrane. While a further signal drop below 0.5 for the control in Fig. 4C is possible, the causes in the long term are most likely more related to degradation of the NBD dye over such long time scales rather than spontaneous lipid flip-flop. We have added a statement on page 13 about the naturally low rate

of lipid flip-flop of POPC in reconstituted vesicles which supports their usage for lipid scrambling experiments and we thank the reviewer for strengthening the message of this work. However, we feel that changing the lipid composition and studies of how lipid scrambling depends on the type of lipids goes beyond the scope of this manuscript. However, we plan to conduct these experiments in the future.

***Question 3:** maybe I missed this point, but when comparing the scramblase efficiency – is it normalized to transferred lipids per pore? per area? Do you really compare the efficiency of a single scramblase with a single DNA nanostructure, or is this an effect of different densities, etc.?*

Response 3: The very useful comment of the reviewer made us aware that the paragraph discussing the scrambling rates of our DNA nanostructure had to be clarified. We have rewritten the entire paragraph in a way highlighting which rates are compared to each other and significantly expanded the discussion and interpretation of the different contributions to the observed experimental rates. We have also included estimations on the influences of lipid diffusion and multiple, simultaneously active scramblases. In the revised manuscript we included a more in-depth discussion on pages 17 and 18 together with the detailed calculations on page 22 in the supplementary information.

***Question 4:** the authors claim (on page 13) that scramblase action by the DNA nanostructure is better because of the larger diameter toroidal lipid pore. This should also be true for other DNA pore structures by the same group and by others. Is there any (e.g., computational) evidence for general scramblase action of all of these structures? Or is the four-helix structure special?*

Response 4: The reviewer is absolutely right: all DNA nanostructures that form toroidal lipid pores should catalyze lipid scrambling. Proving lipid scrambling experimentally, is, however, more difficult for the pores that contain a hollow inner passage as the passage could let dithionite molecules inside the vesicle. We could, however, use the results of our previous MD simulations to estimate the rate of lipid scrambling produced by a single DNA duplex [3], a six-helix nanopore that does not form a toroidal pore [4,5] and a DNA funnel [6]. As expected, the rate of lipid scrambling is approximately proportional to the radius of the toroidal nanopore with ~4 and ~200 lipids/ μ s for the single helix and funnel, respectively. In contrast, no lipid scrambling was observed for a DNA nanostructure that had a modified DNA backbone and did not allow the toroidal pore to form [4,5]. We comment on this observation on page 8 of the revised manuscript.

***Question 5:** in Fig. 4F it would be good to see the results of the negative control experiment also in the main paper.*

Response 5: We have included the negative control experiment now in the main paper under Fig. 4e and f and made appropriate changes to the main text and Supplementary Fig. 17. This

should make the effect of the DNA nanostructure on the cells clearer for the reader, we therefore thank the reviewer for this helpful suggestion.

Question 6: *the title is really a little too catchy (but this is personal taste) – it sounds a little like a newspaper headline.*

Response 6: We amended the title which should now be less like a newspaper headline but still summarizes the main message of the work and attracts the attention of the reader.

Question 7: *it is interesting pointing out that this specific “reaction” of lipid flipping is “catalyzed” much better than by scramblases. Nevertheless, this is a very special reaction – no substrate specificity, no covalent bond formation involved, etc. - and probably the hopes for creating artificial enzymes with similar efficiency for other reactions are rather low. It would therefore be good to differentiate this point a little better within the text.*

Response 7: We thank the reviewer for pointing out that the special case of enzymatic activity induced by our DNA scramblase was not clear enough. We have therefore added a specific comment in our conclusions on page 19 (bottom).

References:

1. Sperotto, M. M. & Ferrarini, A. In *The Biophysics of Cell Membranes: Biological Consequences* (Epanand, R. & Ruysschaert, J.-M. eds.) 29–60 (Springer, 2017).
2. Nakano, M. *et al.* Flip-flop of phospholipids in vesicles: kinetic analysis with time-resolved small-angle neutron scattering. *J. Phys. Chem. B* **113**, 6745–6748 (2009).
3. Göpfrich, K. *et al.* Ion Channels Made from a Single Membrane-Spanning DNA Duplex. *Nano Lett.* **16**, 4665–4669 (2016).
4. Burns, J. R., Stulz, E. & Howorka, S. Self-assembled DNA nanopores that span lipid bilayers. *Nano Lett.* **13**, 2351–2356 (2013).
5. Yoo, J. & Aksimentiev, A. Molecular Dynamics of Membrane-Spanning DNA Channels: Conductance Mechanism, Electro-Osmotic Transport, and Mechanical Gating. *J. Phys. Chem. Lett.* **6**, 4680–4687 (2015).
6. Göpfrich, K. *et al.* Large-Conductance Transmembrane Porin Made from DNA Origami. *ACS Nano* **10**, 8207–8214 (2016).

Reviewer #4 (Remarks to the Author):

Ohmann et al. have developed a synthetic molecular DNA-based complex able to foster the translocation of lipids across lipid membrane and cell membrane structures. Using both computer simulations and experiments they demonstrate the proof of principle that these synthetic molecules can increase the translocation rate more than any known biological (scramblase) protein. The findings are unique and the concept used is novel. Given that lipid translocation is in a central position in many biological processes such as apoptosis, the results discussed in the paper can be considered to be important, and of broad interest.

Response: We thank the reviewer for recognizing a unique and novel character of our work, as well as its potential importance and broad appeal.

Question 2: *Yet, there are several points that the paper does not discuss to a sufficient degree.*

While the findings are appealing, the paper does not provide a convincing explanation as to what exactly happens, and why?

Response 2: We thank the reviewer for alerting us about potential confusions about the mechanism of lipid scrambling. As we explain in detail below, and also now stress in the revised manuscript (bottom of page 5 and page 7), our DNA nanostructure enables passive diffusion of lipids from one leaflet of the membrane to the other by forming a toroidal lipid pore that connects the two leaflets.

Question 3: *The paper shows that the DNA nanostructure explored in the study increases the lipid translocation rate very significantly. However, what exactly happens, how does the translocation process take place? What is the reaction/transition path of lipids during the translocation process? How do the lipids orient themselves during the translocation process, such as, do they for instance expose their polar head group to the DNA nanostructure in a manner that would show electrostatic interactions to predominate the translocation process? Do the lipids translocate one by one, or are there concerted dynamic motions involved, leading to translocation of clusters of lipids in a concerted fashion?*

Response 3: Figures 2c and d, Supplementary Figs. 3 and 4 of the original manuscript, as well as Supplementary Movies 1-4, provide a detailed mechanistic account of the lipid translocation process. That is, Figure 2c shows a sequence of microscopic configuration illustrating spontaneous inter-leaflet transfer of one lipid molecule during a 2.2 microsecond MD simulation whereas Figure 2d plots the radial and z coordinate of that lipid's head group versus simulation time. The plots illustrate reorientation of the lipid molecule as it approaches the DNA nanostructure, which exposes the lipid's polar head group to the DNA nanostructure. To make these observations more explicit, we added arrows to Figure 2c to illustrate the direction of motion and the pathway of the highlighted lipid molecule. We also added a sentence to the main text (page 7) to highlight reorientation of the lipid molecule and passage of the lipid head group near the DNA nanostructure.

Our simulations have also shown that inter-leaflet transport of lipids occurs spontaneously and hence transport of individual lipid molecules is not correlated. In our toroidal pores all lipid molecules that line the inside of the pore perform diffusive motion within the leaflets and thus can be thought of as moving from one leaflet to the other. We comment on this observation on pages 5 and 7 of the revised manuscript.

Question 4: *Moving on, the question why? What are the cause and the driving force for the observed motion? The authors could have used their simulation data for the observed translocation processes along the DNA nanostructure as a basis for free energy profile calculations, and by doing this, they could have computed the free energy profile for translocation across a membrane and determined how it depends on lipid-DNA interactions (in varying positions along the DNA nanostructure surface)? What is the free energy barrier for translocation, where is the largest barrier along the transition path, and what causes it? If the translocation is considered at several temperatures, is it the enthalpic or the entropic component that dominates the translocation process and the free energy barrier that apparently is much lower than the barrier associated with lipid translocation in the absence of the DNA nanostructure?*

Response 4: The driving mechanism of inter-leaflet transport of lipids is diffusion, a random displacement of individual molecules driven by stochastic forces of the environment. We now stress this fact on pages 5 and 7 of the revised manuscript.

We thank the reviewer for suggesting a free-energy calculation. First, we note that our MD simulations already provide extensive sampling of lipid positions within and away from the toroidal pore, which presents a straightforward opportunity to evaluate the free energy change during a lipid passage through Boltzmann inversion. We added Supplementary Fig. 5a that plots the two-dimensional free-energy profile obtained by applying the Boltzmann inversion procedure to the lipid phosphorous atom density averaged over about the azimuthal angle and over the MD trajectory. A corresponding one-dimensional profile through the two-dimensional free energy landscape is now also shown in Supplementary Fig. 5b. Indeed, we find that there is only free energy barrier of $\sim 1 k_B T$, consistent with the rapid scrambling we observed. We comment on the free energy profile calculations on page 7 of the revised manuscript.

Although the question about relative contribution of enthalpic and entropic forces to the free energy barrier may merit further investigation it will not be as simple as repeating our simulations at different temperatures as the temperature will affect not only the DNA-lipid interaction but also the structure of the lipid bilayer. Furthermore, it is not clear to which extent knowing the relative contributions would substantially enhance our proof-of-principle demonstration of a synthetic scramblase made of DNA. Hence, we defer the study of enthalpic and entropic contributions to our future work.

Question 5: *Unfortunately, these two questions (and the many follow-up questions resulting from these two key questions) are not addressed adequately. The paper demonstrates a unique finding, which is valuable, but the paper is not yet able to describe what really happens and why?*

Response 5: As detailed above, we have revised our manuscript to make our description of the mechanism of lipid transport more accessible and clarify the mode of action.

Question 6: *As to more minor questions, the authors are encouraged to consider the following.*

The paper shows that in order to be functional, the DNA nanostructure should have two cholesterol tags attached to it. Why? Why one is not sufficient? Again, calculations of free energy profiles associated with translocation paths could provide a means to explain the matter.

Response 6: We thank the reviewer for pointing out the need to clarify this. In general, insertion of a cholesterol tag lowers the system's free energy whereas creation of a toroidal pore comes with an energy penalty. Insertion of a DNA nanopore becomes energetically favorable when the free-energy gain from moving the cholesterol tags from solution to lipid bilayer environment outweighs the free-energy penalty of a toroidal pore formation [1]. Experimentally, we have shown that insertion of a four-helix DNA bundle into a lipid membrane requires attachment of at least two cholesterol anchors [2]. Another group has recently confirmed that two anchors are necessary with a six-helix DNA structure [3].

Because incorporation of one cholesterol tag comes with the same favorable free energy gain regardless of the lipid configuration, one can naturally expect the lipid bilayer to be in its lowest energy state, which is a planar bilayer with no toroidal pore in it. The case of two cholesterol anchors is more complex, as simultaneous insertion of two anchors may be possible only for non-planar lipid bilayer configurations, including a toroidal nanopore. Although the process of a DNA nanopore insertion deserves a thorough investigation, we reserve it for our future work as the focus of the present study is on demonstration of lipid scrambling catalyzed by nanopore insertion. We added a brief comment referencing to the above discussion of previous findings on page 15 of the revised manuscript.

Question 7: *If the reviewer is not mistaken, the reported translocation rates describe absolute numbers across a membrane (from upper to lower, and from lower to upper leaflet). However, what is the net rate across the membrane? Is it different from zero?*

Response 7: This reviewer is correct: our translocation rate describes the absolute number of inter-leaflet crossing events. Because lipid crossing is facilitated by diffusion, it occurs in both directions. Hence, the average net rate of lipid transfer is close to zero, but it varies stochastically from one simulation to the other. For example, 54% of the lipid crossings observed in the 2 μ s simulation were from the lower leaflet to the upper leaflet, and the remaining 46% move from the upper leaflet to the lower leaflet. We stress this observation in a brief comment on page 7 of the manuscript.

Question 8. While this may be speculative, is there reason to assume, based on the results reported in the paper, that the DNA-mediated translocation rates would be lipid dependent?

Response 8: Yes, our simulations suggest that the rate of lipid transport can depend on the lipid type. Specifically, spontaneous transport of a DPhPE lipids occurred approximately three times slower than the transport of PC lipids (compare Figure 2e and Supplementary Fig. 4f), which we attribute to stronger interaction of the PC lipid head groups with the DNA nanostructure. We comment on this possibility on page 7 of the revised manuscript.

References:

1. Göpfrich, K. et al. Large-Conductance Transmembrane Porin Made from DNA Origami. *ACS Nano* **10**, 8207–8214 (2016).
2. Göpfrich, K. et al. DNA-Tile Structures Induce Ionic Currents through Lipid Membranes. *Nano Lett.* **15**, 3134–3138 (2015).
3. Burns, J. R. & Howorka, S. Defined Bilayer Interactions of DNA Nanopores Revealed with a Nuclease-Based Nanoprobe Strategy. *ACS Nano* (2018). doi:10.1021/acsnano.7b07835

REVIEWERS' COMMENTS:

Reviewer #2 (Remarks to the Author):

The authors have taken remarkable care to address comprehensively all the points that I raised (including the critical point of dithionite permeation through the nanopore) as well as points raised by the other reviewers. This is a comprehensive revision of a very interesting and innovative paper.

Reviewer #3 (Remarks to the Author):

The authors have responded well to all of the referees' comments and also included additional experiments to support their claims.
It is recommended to accept the paper for publication.

Reviewer #4 (Remarks to the Author):

The authors have improved the manuscript significantly. They have clarified the questions presented in the previous report and complemented the manuscript with new data, describing the free energy landscape and molecular-scale mechanisms to a sufficient degree. It is therefore a pleasure to recommend publication of the manuscript.